# Larval development and survival of pond-breeding anurans in an agricultural landscape impacted more by phytoplankton than surrounding habitat

**Melissa B. Youngquist[1,2], Michelle D. Boone[1] ***

**1** Department of Biology, Miami University, Oxford, Ohio, United States of America, **2** Daniel P. Haerther Center for Conservation Research, John G. Shedd Aquarium, Chicago, Illinois, United States of America

* boonemd@miamioh.edu

**Data Availability Statement:** Data is accessible via the link and is now publicly available after acceptance: 10.6084/m9.figshare.14555976 https://figshare.com/articles/dataset/dx_doi_org_10_6084_m9_figshare_6025748/6025748.

## Abstract

The destruction of freshwater habitat is a major contributor to biodiversity loss in aquatic ecosystems. However, created or restored wetlands could partially mitigate aquatic biodiversity loss by increasing the amount of available habitat across a landscape. We investigated the impact of surrounding terrestrial habitat and water quality variables on suitability for two species of pond-breeding amphibians (bullfrogs [*Lithobates catesbeianus*] and Blanchard's cricket frogs [*Acris blanchardi*]) in created permanent wetlands located on an agricultural landscape. We examined tadpole growth and survival in field enclosures placed in ponds surrounded by agricultural, forested, or grassland habitats. We also evaluated the potential for carryover effects of the aquatic environment on terrestrial growth and overwinter survival of cricket frog metamorphs. We found that habitat adjacent to ponds did not predict tadpole growth or survival. Rather, phytoplankton abundance, which showed high variability among ponds within habitat type, was the only predictor of tadpole growth. Cricket frogs emerged larger and earlier from ponds with higher phytoplankton abundance; bullfrogs were also larger and at a more advanced developmental stage in ponds with higher levels of phytoplankton. Overwinter survival of cricket frogs was explained by size at metamorphosis and there were no apparent carryover effects of land use or pond-of-origin on overwinter growth and survival. Our results demonstrate that created ponds in human-dominated landscapes can provide suitable habitat for some anurans, independent of the adjacent terrestrial habitat.

## Introduction

Habitat destruction is the largest driver of biodiversity loss globally and freshwater ecosystems are among the most threatened [1]. Within the continental USA over 50% of wetlands have been lost since 1780, with some midwestern states losing nearly 90% [2]. Overall wetland loss has slowed in the last decade in part because of an increase in created freshwater ponds in agricultural, urban, and grassland habitats [3], which exceeds 2.6 million in number and accounts

**Funding:** Funding from ASG (Amphibian Specialist Group)/ARMI Seed Grant. The funders had no role in study design, data collection and analysis, decision to publish, or preparation of the manuscript.

**Competing interests:** The authors have declared no competing interests exist.

for roughly 20% of standing water in the USA [4]. In areas where wetland loss has been extensive, created ponds are frequently the only freshwater habitat available [3]. In contrast to the variable hydroperiods and extensive littoral zones of natural wetlands, many created ponds have permanent hydroperiods with a narrow region of shallow, near-shore habitat [3, 5]. While the ability of created wetlands to replace specific ecosystem services of natural wetlands has varying outcomes (e.g., [6, 7]), created wetlands can provide effective disturbance control, commodities and recreation, and production of biodiversity [8]. Because a variety of aquatic vertebrate and invertebrate taxa are known to utilize created ponds [9–11], these habitats can be important for maintaining freshwater diversity within anthropogenic landscapes. Determining the relative value of aquatic systems is of paramount importance, particularly for taxa like amphibians that are experiencing global population declines from habitat loss and other factors [12].

Amphibians with complex life cycles depend on freshwater habitats for reproduction. While many use wetlands with temporary and semi-permanent hydroperiods, they also use created wetlands and ponds with permanent hydroperiods [11, 13, 14]. Certain features like shallow water access, emergent vegetation, and absence of fish are particularly important in determining amphibian species richness and juvenile recruitment in created wetlands [15]; however, the relative importance of the surrounding landscape for larval development is less well studied. For created ponds in agricultural landscapes, surface waters can be affected by contaminants and excess nutrients from nonpoint inputs, leading to eutrophication that can be traced to nutrient runoff [16, 17]. Therefore, created ponds adjacent to agricultural fields may have higher levels of pesticide and fertilizer contamination than ponds in forests or grasslands, which may be less impacted by runoff from anthropogenic sources. The quality of created ponds may be further affected by the presence or lack of natural terrestrial buffers, which help protect water quality by filtering silt and contaminants from runoff (reviewed in [18, 19]). Management decisions like mowing near ponds, therefore, can influence runoff and thus larval amphibians [20]. Terrestrial buffers also provide vital habitat for juvenile and adult life stages [21]. Because larval amphibians can be sensitive to changes in food resources and water quality metrics that are affected by the terrestrial environment (e.g., [22, 23]), we need to evaluate the role of surrounding land use on recruitment of pond-breeding amphibians in created ponds to direct effective management and to determine ways of improving water quality.

The primary objective of our study was to determine how terrestrial habitat surrounding permanent created ponds and water quality of those habitats impacts tadpole growth and survival. Our secondary objective was to assess whether there were carryover effects of the larval environment on juvenile growth and survival through overwintering. The importance of the larval environment for adult growth, survival, and overall fitness has been well documented across taxa (e.g., [24]). For amphibians, size at metamorphosis, which is often affected by aquatic conditions (i.e., predators, competitors, and/or contaminants), can affect juvenile survival [25–27].

We reared two amphibian species, bullfrogs (*Lithobates catesbeianus*) and Blanchard's cricket frogs (*Acris blanchardi*; hereafter, cricket frogs), in created wetlands found in three common land use types within anthropogenic landscapes: row-crop agricultural land, restored grassland, and deciduous forest. We chose these two species because they breed in permanent wetlands within all three habitats [13, 15, 28, 29] and have shown sensitivity to aquatic conditions [30–32]. Despite these similarities, bullfrogs and cricket frogs differ in their conservation status. Bullfrog populations are stable within their range and the species is considered invasive in countries around the world [33], often flourishing in contaminated sites [34]. In contrast, cricket frogs are declining at the edges of their range [28, 29] for reasons that are unclear. Understanding how land cover surrounding created ponds affects juvenile recruitment could

provide insight into why cricket frogs are declining while bullfrogs are stable and expanding in agriculturally dominated areas. We hypothesized that tadpole growth and survival would be greatest in more natural habitats and that species would differ in their sensitivity to surrounding pond habitat; we predicted that cricket frogs would be more sensitive to surrounding habitat than bullfrogs. We also hypothesized that the effects of aquatic habitat on larval life stages would carryover and affect terrestrial growth and overwinter survival.

## Materials and methods

### Ethics statement

The office of Research Ethics and Integrity Program at Miami University reviewed and approved our study protocol through the Institutional Animal Care and Use Committee (IACUC) via approval number IACUC 827. Animal collection was approved by Ohio Department of Natural Resources (Wild Animal Permit: Scientific Collection 14–80). Individuals that reached metamorphosis or were tadpoles at the end of the field enclosure study (Experiment 1 below) or that were captured in enclosures at the end of terrestrial study (Experiment 2 below) were euthanized using MS-222, which was required by our animal collection permit from the Ohio Department of Natural Resources. Other individuals are presumed to have died in the field under natural conditions. Approximately 550 tadpoles of cricket frogs and 550 tadpoles of bullfrogs were added to a total of 54 enclosures across ponds at field sites; 126 metamorphosed cricket frogs from the field enclosures were subsequently reared in terrestrial enclosures through overwintering (described below).

### Animal collection and care

We collected five amplectant cricket frog pairs on 19 May 2013 and four partial bullfrog egg masses on 20 and 21 May 2013 from Miami University's Ecology Research Center (ERC) in Oxford, OH (Butler County). The pond at the ERC is similar to the experimental sites for this study; it is a permanent created pond, has a partially mown grassy buffer, and is open canopy. Amplectant cricket frogs were held in the lab overnight in plastic shoebox containers with 3 cm of water and multiple twigs. Eggs were mixed within species and placed in outdoor mesocosms that were set up on 14 May 2013 containing 220 L aged city water (depth of 10 cm), leaf litter, and algal inoculate from a fishless pond. Tadpoles were maintained in mesocosms until addition to in situ enclosures.

### Experiment 1: Effects of habitat on tadpole performance

To test for effects of upland habitat on anuran growth and survival we selected three replicate constructed permanent ponds in each of three habitat types (agriculture, forest, and grassland) for a total of nine ponds (see Fig 1 for examples of pond types). Ponds were randomly selected from a dataset of ponds in a 10 x 20 km area within Butler and Preble Counties, OH; these particular ponds were used because we were able to obtain land-owner permission. The number of pond replicates used for this study was limited by land-owner permission and logistics of the experiment; however, other studies have found significant habitat effects using similar design and replicate size (e.g., [14, 20, 35]). All ponds used in this study were used for irrigation and/or recreation, but there were no observed drawdowns in any of the ponds during the study. Ponds were of similar size (mean [SE] = 0.433 [0.126] ha) and all had populations of centrarchid fishes that were added to ponds by landowners in the past. Two ponds were at least 13 years old (one agriculture, one grassland) and the rest were at least 19 years old [36, 37]. Bullfrogs and cricket frogs are commonly found in ponds with fish [13] and there was

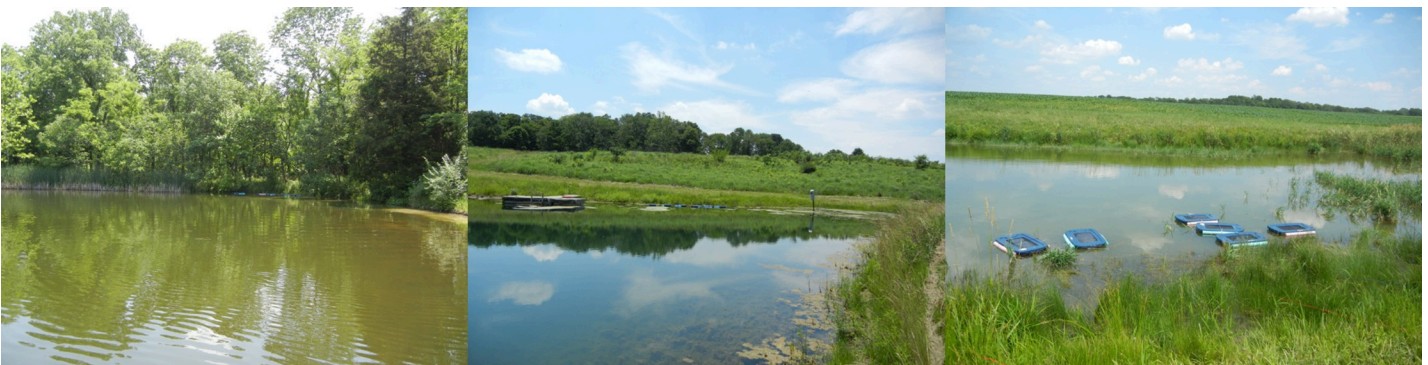

**Fig 1. Three of the field study sites.** From left to right: a forested, grassland, and agriculture site, with enclosures visible in the agricultural site.

evidence of successful breeding by either bullfrogs, cricket frogs, or both at each of our study sites, indicating that these ponds were representative of those used by these species.

Agricultural ponds were surrounded by a grassy buffer ≤50 m from agricultural plots; crops in agricultural plots included a mix of corn and soybean, soybean only, or a mix of soybean and sunflower. The grassland sites were in the Conservation Reserve Program (CRP) and had been out of agricultural production for at least eight years; one grassland pond had 1/3 of the pond adjacent to mixed deciduous forest. The enclosures in this pond were placed on the grassy side of the pond and received full sun. The forested ponds were in mixed, deciduous forest: one pond was in an old pine plantation undergoing succession such that half the pond was adjacent to primarily pine trees; a second pond had 1/3 open to a grassland undergoing succession, while the other 2/3 of the pond was adjacent to disturbed remnant forest that contained high abundance of amur honeysuckle (*Lonicera maackii*), an invasive shrub; and the third pond was surrounded by mature deciduous forest. Enclosures in the forested ponds were placed on the side that was adjacent to mature deciduous trees.

We added six enclosures to each pond–three replicate enclosures per species. Enclosures were constructed out of 113 L plastic totes (72 x 36 x 43 cm; L x W x H). We cut out the sides and covered the openings with 2 mm mesh screen. We covered the tops with a screen lid to allow for light penetration. Enclosures were floated in ponds with the top approximately 10 cm out of the water (~85 L water) and secured to rebar posts. We added 100 g of mixed deciduous leaf litter to each enclosure. Enclosures were installed on 3 and 4 June 2013; 20 cricket frog or bullfrog tadpoles (Gosner stage 25 [38]) were added to each enclosure in each pond on 5 June 2013 (after 15–16 days in outdoor mesocosm ponds as eggs and tadpoles). Three enclosures across two forest ponds (rep 1 [1 bullfrog and 1 cricket frog enclosure] and rep 3 [1 bullfrog enclosure]) failed in the first two weeks (two tipped over prior to being secured to rebar posts on 14 June 2013; one had mesh siding that became unglued). We replaced tadpoles in these three enclosures on 17 June 2013 with animals from the same clutches that were still at Gosner stage 25–26 (day 12, after being held 28 days in outdoor mesocosm ponds as eggs and tadpoles).

We collected pond-level data on dissolved oxygen (DO), pH, phytoplankton, periphyton, dissolved nitrogen, dissolved phosphorus, and suspended solids on experimental days 6, 21, 35, and 62. We collected periphyton by scraping 27.5 cm$^2$ from microscope slides, which had been suspended in the water column since installation of the enclosures in the ponds. Periphyton was placed onto 47 mm 0.7 μm glass fiber filters, which were submerged into 15 mL of buffered acetone solution. Phytoplankton was collected by vacuum filtering 100 mL of water taken from a 3 L composite water sample onto a 47 mm 0.7 μm glass fiber filter that was placed

in 15 mL of buffered acetone solution. Periphyton and phytoplankton samples were stored overnight at 4˚C for chlorophyll extraction. We estimated phytoplankton and periphyton relative abundance from chlorophyll *a* concentration via fluorometry (10-AU fluorometer, Turner Designs). For suspended solids, we vacuum filtered pond water through 47 mm 0.7 μm glass fiber filters, dried samples for 24 hours at 60˚C, and calculated mg dry mass $L^{-1}$. For dissolved nutrient analyses (nitrate and phosphorus), we reserved 80 mL of filtrate after phytoplankton and suspended solid extraction. We added 120 μL of $H_2SO_4$ to the filtrate and refrigerated samples until analysis (QC 8000 FIA autoanalyzer, Lachat Instruments). Dissolved nitrogen was analyzed using a standard cadmium reduction method and total dissolved phosphorus was analyzed using an orthophosphate method. Enclosures were monitored twice weekly prior to cricket frog metamorphosis and every day thereafter. Cricket frog metamorphs were collected after Gosner stage 42 [38] and held in the lab until tail resorption (less than 1 week). We terminated the experiment on 12 August 2013 (day 68), at which time we collected all remaining cricket frog and bullfrog tadpoles. We believed most cricket frogs had metamorphosed, and we collected fewer than 1–2 from any enclosure except for one forest pond, where we collected an average of 9 tadpoles per enclosure at the end of the study; cricket frog tadpoles that did not metamorphose by the end of the study were not included in analyses.

Cricket frog response variables were survival to metamorphosis, time to metamorphosis, and mass at metamorphosis. Because bullfrogs do not often reach metamorphosis in a single season, bullfrog response variables were tadpole survival, tadpole mass, and Gosner developmental stage [38]. Bullfrog and cricket frog responses were analyzed separately. Our experimental unit was the enclosure. Mass was log transformed for both species to meet assumption of normality. First, we tested for effects of habitat using generalized linear mixed models for survival (binomial distribution) or linear mixed effects models for variables with a Gaussian distribution (mass or Gosner developmental stage); habitat was a fixed effect and pond was a random effect. For bullfrog survival, there was complete separation and we had to drop the random variable; therefore, we used a generalized linear model to test for effects of habitat on bullfrog survival. We used analysis of variance (ANOVA) to evaluate whether habitat had a significant effect on any response variable; we used Chi-square test statistic for survival data and the Kenward-Roger method of approximating the F test statistic for growth and development responses. Second, we used analysis of variance (ANOVA) to test for differences between ponds. We dropped one agricultural pond from the analysis because of 100% mortality, likely caused by predatory backswimmers (Notonectidae), which colonized the enclosures at extremely high abundances within the first couple weeks. However, this pond also had phytoplankton (1623.2 ± 678.3 [SE] μg/L) and phosphorus (7752.3 ± 2434.4 [SE] μg/L) levels that were one or two orders of magnitude higher, respectively, than all other ponds (see results below), so we cannot rule out the effects of eutrophication on tadpole survival. Inclusion of this site did not alter results regarding effects of habitat on larval survival.

To examine effects of water quality across habitat types, we used linear mixed effects models and the Kenward-Roger method of approximating the F-test to test for effects of time and habitat; we used pond as a random effect. We then averaged each water quality variable across time to get a single value per variable per pond to test for water quality correlations with tadpole growth and development (mass, time to metamorphosis, or stage) using non-parametric rank regressions. We tested each water quality variable separately to prevent overfitting the data because of small sample size (8 pond sites). We tested phytoplankton, periphyton, dissolved nitrogen, pH, and dissolved oxygen. We corrected for multiple comparison by using one-stage false discovery rate-adjusted p-value (*FDR-adjusted P*; [39]). Phytoplankton, dissolved phosphorus, and suspended solids were highly correlated (r > 0.9), so we excluded

dissolved phosphorus and suspended solids from analysis. Phytoplankton was log-transformed to improve model fit. Analyses were conducted in the R statistical platform [40].

## Experiment 2: Carryover effects on cricket frog growth and overwinter survival

Following the field study, we used a common garden experiment to examine possible carry-over effects of the larval environment on cricket frog terrestrial growth and overwinter survival. We reared cricket frog metamorphs in 2 x 2 m outdoor enclosures located in a grassy field. We grouped cricket frogs from the same pond together in terrestrial enclosures in a nested design, although we used only two ponds from each habitat type because of low survival in some ponds. Each larval treatment was replicated three times in terrestrial enclosures (i.e., 3 larval habitat types X 2 ponds from each habitat type X 3 terrestrial replicates = 18 terrestrial enclosures). Each enclosure had a ~0.5 m deep and ~0.5 m wide pit in its center, filled with leaves from a mixed deciduous forest, as an overwinter refugium. Wire mesh baffles were placed over the tops of each wall to prevent escape. We added seven cricket frog metamorphs to each enclosure. Cricket frogs were given a unique toe clip ID and added to enclosures immediately after metamorphosis. Because cricket frogs are semi-aquatic, remaining near aquatic habitats though the summer, they are prone to desiccation stress when held in terrestrial enclosures. Therefore, we watered enclosures with a sprinkler about one day each week (~4 hours per day) in July and August to minimize desiccation stress. In the event of natural precipitation, we would not water enclosures that week. We collected cricket frogs on 5 April 2014 to record overwinter growth and survival. We checked terrestrial enclosures weekly until no more cricket frogs were encountered (19 April 2014), suggesting that we had captured most to all cricket frogs that survived overwintering.

We used mixed models to examine effects of habitat type and linear models to test effects of pond origin on overwinter survival and terrestrial growth (final mass and change in mass). For mixed models, we used habitat as a fixed effect, pond as a random effect, and terrestrial enclosures were the experimental unit using enclosure means. We used generalized linear models to examine effects of only initial mass on growth and overwinter survival using a Gaussian and binomial distribution, respectively, using individuals as the experimental unit to examine how an individual's size influenced survival and size at the end of the study.

## Results

### Experiment 1: Effects of habitat on tadpole performance

Overall, there were no effects of habitat type (agriculture, field, or forest) on bullfrog or cricket frog growth and survival (Table 1), and survival of neither species was predicted by water quality metrics. There were also no overall differences in any of the water quality metrics among habitat types over time ($F_{6,15} < 2.21$, $P > 0.147$; Mean (Standard Deviation) for 8 ponds used in analyses: pH 8.62 (0.33); DO 9.4 (0.9) mg/L; suspended solids 8.65 (7.22) mg/L; periphyton 5.43 (6.3) µg/L; phytoplankton 57.9 (66.4) µg/L; phosphorus 35.0 µg/L (16.3); nitrate 0.69 (1.8) mg/L). However, there were significant differences in survival for both species among ponds (Table 1; Fig 2). Furthermore, ponds with higher bullfrog growth and survival also had higher cricket frog growth and survival (Figs 2 and 3). After correcting for multiple comparisons, average relative phytoplankton abundance was positively related with cricket frog mass (*FDR-adjusted P* = 0.003) and negatively related to cricket frog time to metamorphosis (*FDR-adjusted P* = 0.009). While not significant after correcting for multiple comparisons, we observed a trend (Table 1) whereby phytoplankton was also positively related with bullfrog

**Table 1. Effects of habitat and pond on bullfrog tadpole and cricket frog metamorphic responses.** Results of ANOVAs and GLMMs to examine the effects of habitat type on bullfrogs and cricket frogs, using pond as a random effect. Stage is Gosner developmental stage and TTM is time to metamorphosis. Degrees of freedom (df) include numerator and denominator df. Test statistic is Chi-square for survival and Kenward-Roger F for stage, mass, and TTM.

| Species | Treatment | Response | df | Test | P |
|---|---|---|---|---|---|
| Bullfrog | Habitat | Survival | 2 | 1.12 | 0.571 |
| | | Stage | 2,5 | 0.97 | 0.441 |
| | | Mass | 2,5 | 0.52 | 0.627 |
| | **Pond** | **Survival** | **7,16** | **7.93** | **0.0003** |
| | | **Stage** | **7,13** | **25.86** | **<0.0001** |
| | | **Mass** | **7,13** | **21.06** | **<0.0001** |
| Cricket Frog | Habitat | Survival | 2 | 0.01 | 0.994 |
| | | TTM | 2,5 | 0.84 | 0.484 |
| | | Mass | 2,5 | 0.52 | 0.603 |
| | **Pond** | **Survival** | **7,16** | **12.11** | **<0.0001** |
| | | **TTM** | **7,15** | **86.07** | **<0.0001** |
| | | **Mass** | **7,15** | **12.8** | **<0.0001** |

Gosner developmental stage (*FDR-adjusted P* = 0.125) and bullfrog mass (*FDR-adjusted P* = 0.090). Tadpoles grew larger and were more developed in ponds with higher amounts of phytoplankton (Fig 3). None of the other water quality variables was significantly correlated with cricket frog or bullfrog growth and development (Table 2).

## Experiment 2: Carryover effects on cricket frog growth and overwinter survival

There were no carryover effects resulting from habitat or pond for cricket frog overwinter survival (mean [SE]: 32 [6] %; Fig 4), final mass (0.83 [0.06] g), or change in mass (0.63 [0.05] g; Table 3). Initial mass, however, was predictive of survival (Chi-sq = 6.64, *P* = 0.010; Fig 5), final mass ($F_{1,46}$ = 12.82, *P* = 0.0008), and change in mass ($F_{1,46}$ = 4.20, *P* = 0.0462). Individuals that metamorphosed larger were more likely to survive and were larger at spring emergence than individuals with a small size at metamorphosis. Cricket frogs that survived had a mean (SE) mass at metamorphosis of 0.20 (0.01) g and those that did not survive had a mean mass of 0.16 (0.01) g; only individuals greater than 0.12 g at metamorphosis survived.

## Discussion

Created ponds have the potential to ameliorate the loss of wetland habitats that contributes to biodiversity loss and may help maintain populations in human-dominated ecological systems. Our study demonstrated that created ponds in three different habitat types served as viable aquatic habitat for larval development of cricket frogs and bullfrogs. We did not detect impacts of adjacent land use on tadpole performance or on water quality; rather, individual pond characteristics varied without respect to immediate surrounding terrestrial habitat type and affected responses of both species similarly despite differences in their conservation status. Our results indicated that resource availability (i.e., phytoplankton) was most important for predicting juvenile recruitment and that resources can be highly variable within a given land use type.

Although land cover can influence pond quality and amphibian communities (e.g., [41, 42]), we did not find that presence of agriculture near the pond had a disproportionate impact relative to forest or grassland; other studies have also found that agricultural ponds are suitable for a variety of anuran species [43, 44]. Yet, the presence of mown and unmown grassy buffers

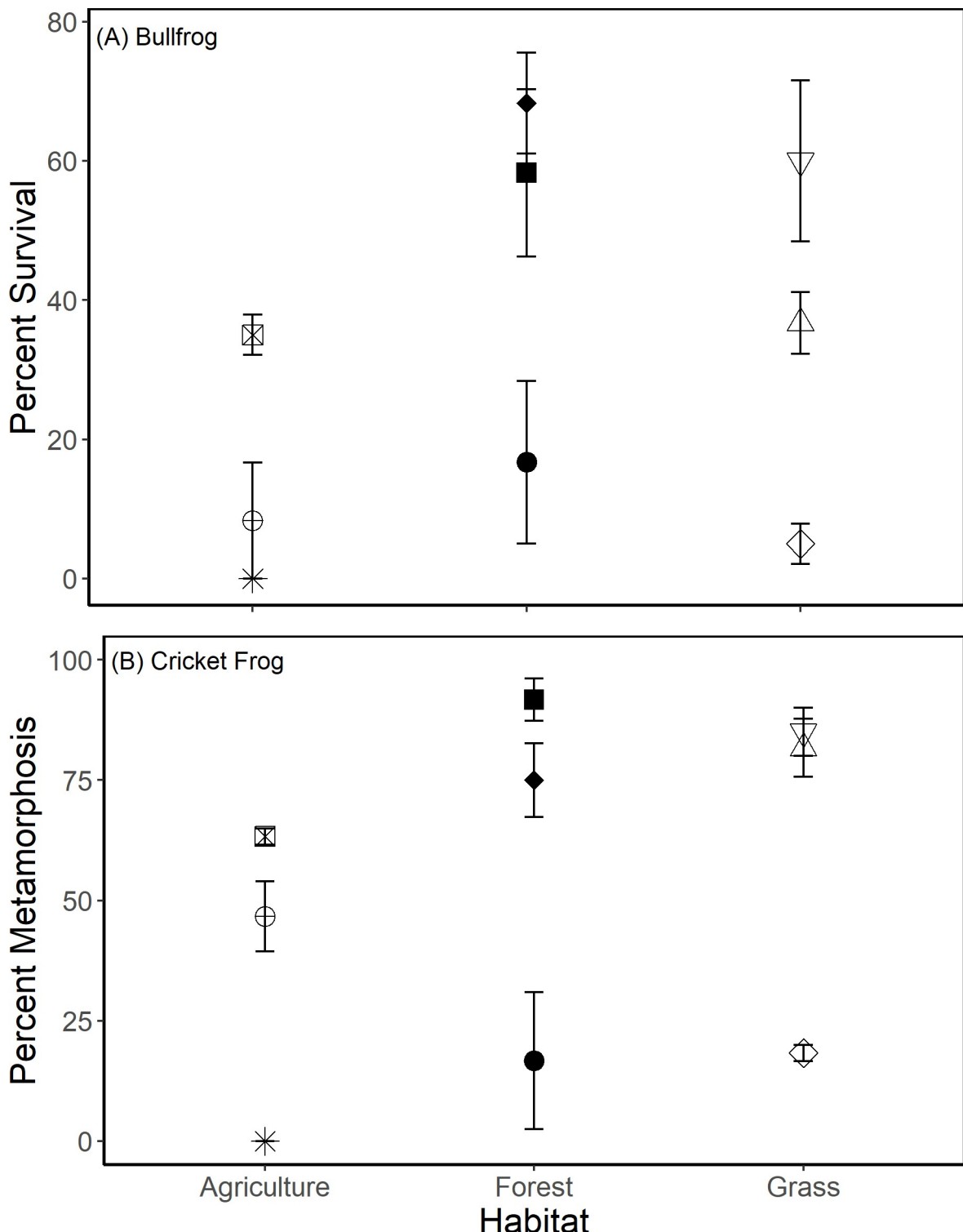

**Fig 2.** Survival within in situ enclosures in ponds for (A) bullfrog tadpoles and (B) cricket frog metamorphs. Error bars are standard errors (SE). Individual points are pond sites; shapes/shading denote individual ponds within a habitat type and are consistent in Figs 2–4 for pond identification.

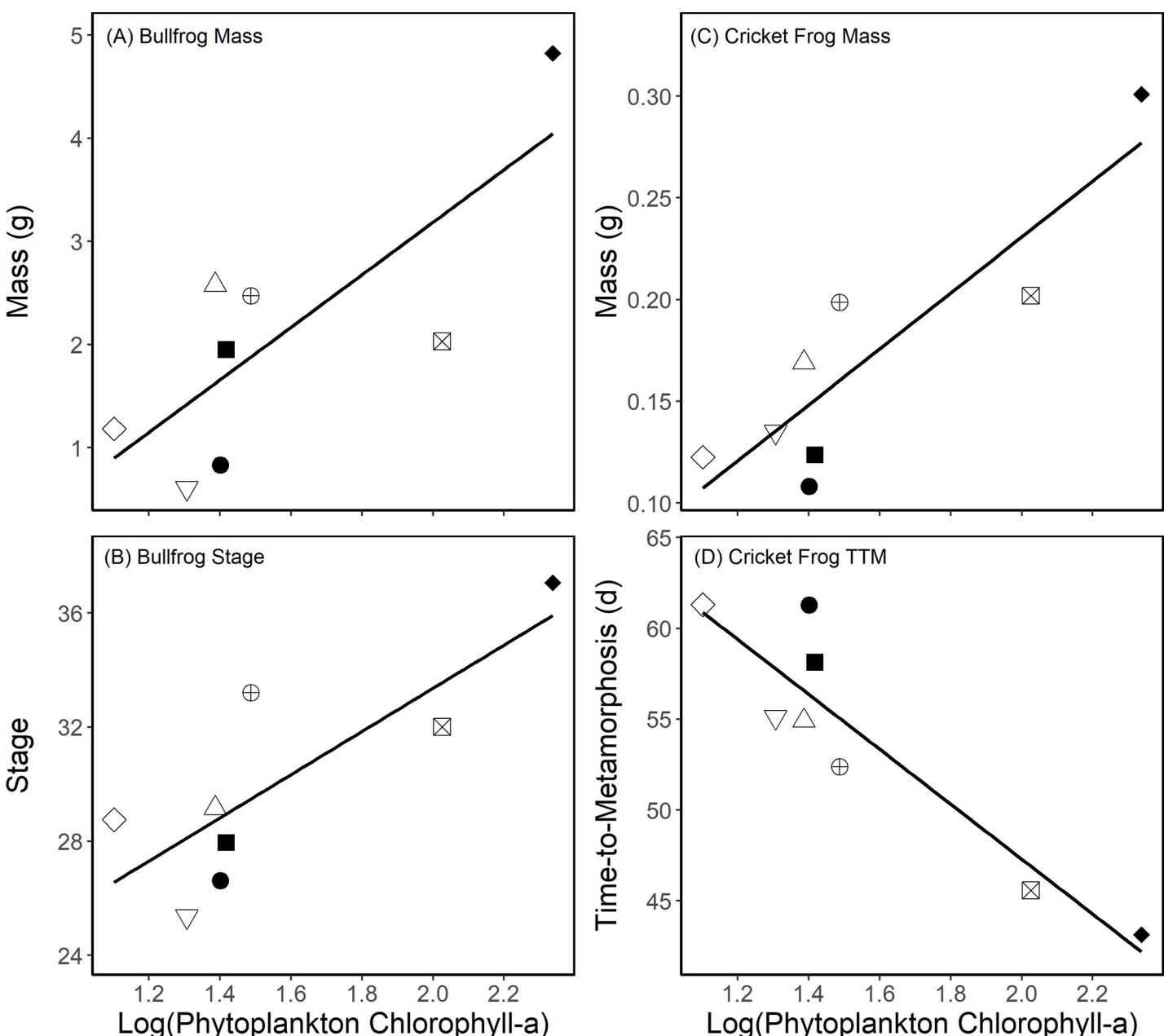

**Fig 3.** Relationship between relative phytoplankton abundance and (A) bullfrog tadpole mass, (B) bullfrog developmental stage, (C) cricket frog mass at metamorphosis, and (D) cricket frog time to metamorphosis. Individual points are pond sites; shapes/shading denote individual ponds within a habitat type and are consistent in Figs 2–4 for pond identification.

around agricultural ponds can filter runoff [20, 45] and could reduce differences between natural and agricultural land uses. The majority of agricultural ponds in this landscape have grassy buffers (MBY, personal observations) and, therefore, our sites and results are likely representative of created ponds in agricultural areas with vegetative buffers. Overall, our sites showed high variation within habitat categories: the ponds were of different ages, cultivated crops were different in each of the agricultural land use types, and two of the non-agricultural ponds were in mixed habitat. All of these factors could have diluted the main effect of

**Table 2. Linear regression between bullfrog and cricket frog growth and water quality metrics.** Stage is Gosner developmental stage and TTM is time to metamorphosis. Bold values are significant after correcting for multiple comparisons with false discovery rate (FDR) adjusted p-values. Italicized values indicate significance before correcting for multiple comparisons.

| Species | Response | Parameter | Parameter Estimate (β) | P | FDR-P |
|---|---|---|---|---|---|
| Bullfrog | Stage | *Phytoplankton* | *8.29* | *0.025* | *0.125* |
| | | Periphyton | −0.347 | 0.355 | 0.888 |
| | | Nitrogen | 0.606 | 0.651 | 0.939 |
| | | pH | −1.44 | 0.939 | 0.939 |
| | | Dissolved Oxygen | 0.996 | 0.864 | 0.939 |
| | Mass | *Phytoplankton* | *2.76* | *0.018* | *0.090* |
| | | Periphyton | −0.173 | 0.158 | 0.395 |
| | | Nitrogen | 0.015 | 0.997 | 0.997 |
| | | pH | −1.23 | 0.583 | 0.972 |
| | | Dissolved Oxygen | −0.249 | 0.860 | 0.997 |
| Cricket Frog | TTM | **Phytoplankton** | **−14.7** | **0.0018** | **0.009** |
| | | Periphyton | −0.003 | 0.999 | 0.585 |
| | | Nitrogen | −1.77 | 0.234 | 0.999 |
| | | pH | 3.81 | 0.861 | 0.999 |
| | | Dissolved Oxygen | 0.252 | 0.996 | 0.999 |
| | Mass | **Phytoplankton** | **0.139** | **0.006** | **0.003** |
| | | Periphyton | −0.002 | 0.842 | 0.983 |
| | | Nitrogen | 0.0124 | 0.393 | 0.999 |
| | | pH | −0.041 | 0.827 | 0.999 |
| | | Dissolved Oxygen | −0.001 | 0.999 | 0.999 |

"habitat." This site-level variability could reduce the likelihood of finding habitat differences if they existed with a small sample size (2–3 ponds per habitat type); yet, similar studies have found effects with 2–3 replicates [14, 20]. If the effects of surrounding land-use/land cover were large we would likely have detected an effect in tadpole responses.

The variability in the land use adjacent to created ponds in this study represent the natural variation found within the region. Our design may not allow strong conclusions regarding narrowly defined land use types, but we can conclude that many created ponds in different land use contexts can provide suitable larval habitat. This conclusion is supported by a survey of cricket frogs from the same region that showed little to no effect of land use type on cricket frog presence [46]. Overall, surrounding habitat alone may not indicate aquatic condition or quality, and food availability in the pond may be the central factor influencing successful survival and development.

The strongest predictor of tadpole success was phytoplankton abundance—a food resource of anuran tadpoles. Bullfrogs and cricket frogs showed similar responses among sites, and we did not find cricket frogs to be more sensitive to land use/land change, as we had expected, than bullfrogs. Ponds that were good for bullfrogs were even better for cricket frogs and both species did poorly in the same ponds. Phytoplankton abundance, which was highly correlated with dissolved phosphorus and suspended solids, predicted tadpole growth for cricket frogs, while bullfrogs appeared less sensitive to phytoplankton abundance; phytoplankton abundance can be positively affected by both increased light availability and nutrient abundance [47]. Adjacent land use, however, did not predict phosphorus, suspended solid, or phytoplankton abundance. The observed variability of water quality within land use type in this study is similar to other studies that have found high variability in the relationship between land use, nutrient loading, and primary production in lakes [48–50].

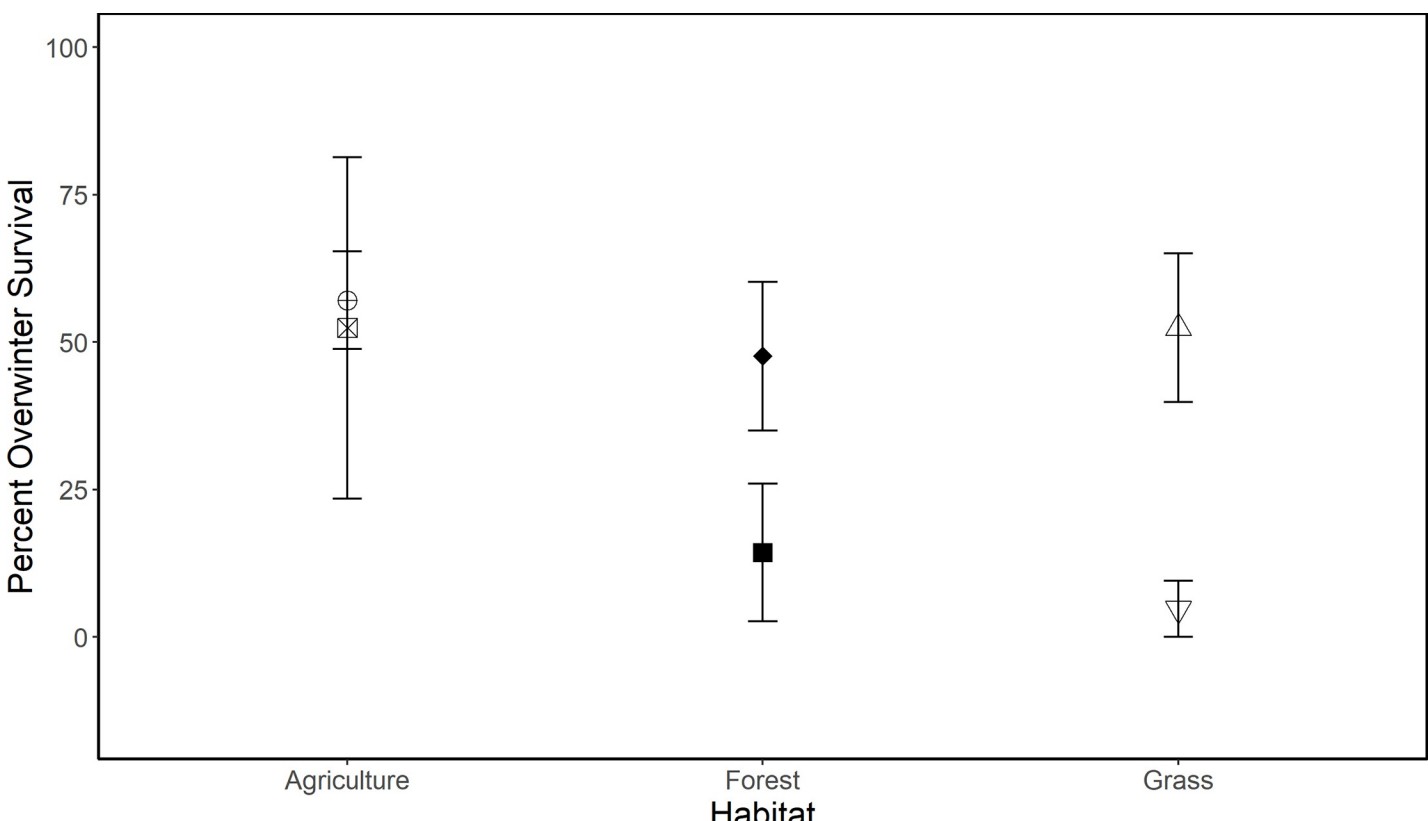

**Fig 4. Cricket frog overwinter survival.** Error bars are SE. Individual points are pond sites that cricket frogs originated from; shape and shading is consistent in Figs 2–4 for pond identification.

Our results support the general observation that algal abundance directly affects the growth of tadpoles, including cricket frog and bullfrog tadpoles [30, 47, 51]. While the tadpoles of both species are food generalists, they generally feed on phytoplankton and periphyton [30, 52, 53]. Our results suggest that high phosphorus concentrations likely increased phytoplankton abundance, which served as the primary food resource for tadpoles. We provided all tadpoles with mixed-deciduous litter for refugia and all tadpoles were caged, both of which could have contributed to the lack of habitat effect on tadpole growth and development; if tadpoles were reared within the physical structure of the pond, the outcome could have been different. Further, detrital inputs are tied to land use and litter type can affect anuran larvae [23, 54]. If litter type was a strong determinant of habitat quality in these ponds we might expect greater algal

**Table 3. Mixed model results for effects of habitat and pond on overwintered cricket frogs.** Degrees of freedom (df) include numerator and denominator df.

| Treatment | Response | *df* | *F* | *P* |
|---|---|---|---|---|
| Habitat | Survival | 2,3 | 0.67 | 0.576 |
| | Final Mass | 2,2 | 0.58 | 0.649 |
| | Growth | 2,2 | 0.36 | 0.742 |
| Pond | Survival | 5,12 | 2.0 | 0.151 |
| | Final Mass | 5,7 | 0.73 | 0.624 |
| | Growth | 5,7 | 0.31 | 0.890 |

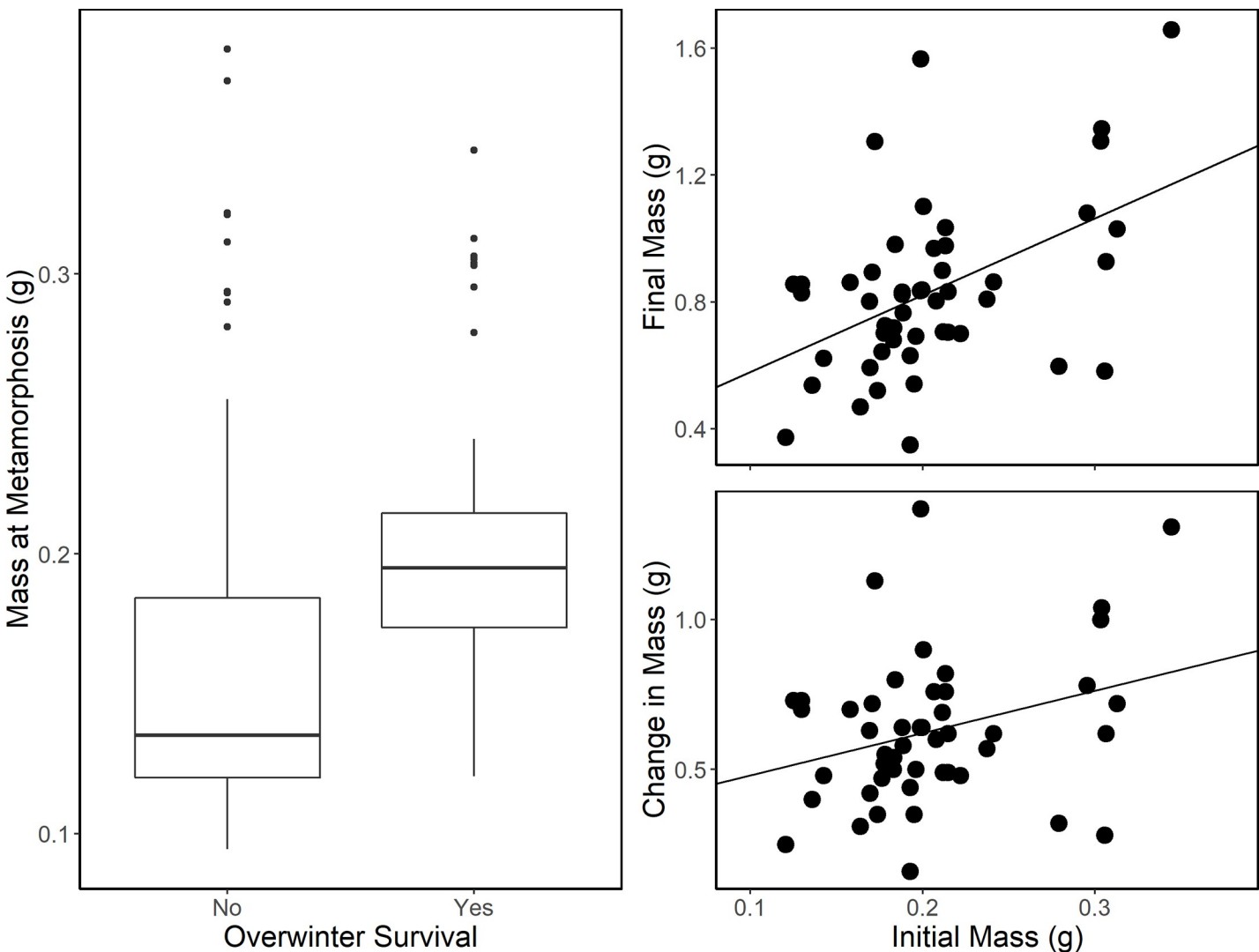

**Fig 5. Cricket frog overwinter survival based on mass at metamorphosis and the relationship between initial mass versus final mass and change in mass of cricket frogs surviving overwintering.** Points represent individual frogs.

growth in grassland ponds, where relatively nutritious grass was the primary litter source [55]. Overall, the conditions in the pond were the most important to anuran growth, regardless of the surrounding terrestrial habitat.

While bullfrogs and cricket frogs successfully transformed across all three habitat types, some environments (e.g., agricultural ponds) may have resulted in pesticide exposure that could impact individuals later in life. Using a common garden experiment for cricket frogs, which complete metamorphosis in a single season, we did not detect any carryover effects of the larval environment from adjacent habitat or individual pond on cricket frog terrestrial growth or overwinter survival. However, size at metamorphosis was an important predictor of survival and growth (similar to [25, 27, 56, 57]); cricket frogs that were larger at metamorphosis were more likely to survive and emerge larger after overwintering. The lack of an overall pond effect, despite differences in size at metamorphosis between ponds (including the subset from each of two ponds added to terrestrial enclosures), was surprising. Yet, there was overlap in the distribution of initial metamorph sizes among ponds. Our data show that cricket frogs

less than 0.12 g at metamorphosis (the smallest individual surviving overwintering) were unlikely to survive the winter. Some species can compensate for small size at metamorphosis or can at least reach larger sizes prior to overwintering (e.g., American toads, *Anaxyrus americanus*; [58]), which could partially explain the lack of a carryover effect from ponds. Because population dynamics are highly sensitive to juvenile survival and condition [59, 60], systems with high food abundance should produce high quality individuals at large sizes (as in [47]), which would be better able to tolerate and survive stressful terrestrial conditions, and these populations would be more likely to persist.

The interaction between larval and terrestrial factors can make carryover effects difficult to predict, although carryover effects on overwinter growth and survival have often not been found in similar studies (e.g., [26, 57, 58]). However, James and Semlitsch [27] found that size at metamorphosis was more important for survival in stressful environments, while carryover effects of sublethal larval exposure to cadmium affected survival in less stressful conditions. For this study, watering enclosures, which made the terrestrial environment more favorable, may have affected the likelihood of detecting carryover effects resulting from factors other than size at metamorphosis. Because our observed survival rates match field observations of cricket frog overwinter survival [61], we infer that watering once a week did not result in unrealistic levels of survival, but it still may have impacted the likelihood of sublethal carryover effects.

Within the midwestern United States, where wetlands loss has been great [4], created grassland ponds (e.g., CRP) can be valuable for amphibian conservation. Yet agricultural ponds too appear to be similarly useful for larval development. Our study demonstrates that created ponds, across land use types, can provide suitable habitat to sustain amphibian larvae though metamorphosis. Further, similar responses of both species suggest that created ponds in agricultural landscapes are unlikely to be an immediate cause of cricket frog declines in Ohio. Certainly, created ponds may be especially beneficial for species that can co-exist with fish, similar to bullfrogs and cricket frogs tested in this study [62, 63]. Anurans in our study were also protected from fish predation within enclosures, so other features of the pond (e.g., shallows, aquatic vegetation; [5]) may influence the likelihood of juvenile recruitment even for fish-tolerant amphibians and could change the success of these anurans under more realistic conditions. Species that prefer temporary hydroperiods and that have no defense against fish will likely be less tolerant of conditions in created ponds like the ones tested in this study.

By investigating only the aquatic life stage in situ (while still examining carryover effects in a common garden experiment), we could not determine if field sites permitted successful terrestrial growth and survival—a necessary sequel for population persistence. Terrestrial buffers around wetlands are vital for juvenile and adult life stages of pond breeding amphibians (e.g., [21]) and may determine if ponds where successful larval development occurs end up functioning as sinks or sources. The landscape context and land use around the pond may be more important for juvenile growth and survival and will likely play an important role in population connectivity. Bullfrogs and cricket frogs, which spend most of the year near aquatic habitat and which can disperse through grassland and crop land [46, 64], may be more tolerant of agricultural landscapes and better able to utilize the created wetlands within these habitat types than species that require undisturbed upland habitats [15, 46]. Thus, understanding how land use affects multiple life stages is needed before we can fully assess the contribution of created wetlands within different land use types across a broad suite of habitat conditions and land use contexts.

Human domination of land ostensibly means that less habitat remains for native wildlife. Yet, created habitats can in some cases restore ecosystems to the landscape in ways that help preserve regional biodiversity [8]. Our study shows that created wetlands in an agriculturally-

dominated landscape show a wide range of heterogeneity in water quality, which influences two native anurans—one invasive and one declining in parts of its range—in a similar fashion. While our study explored the aquatic life stage in these ponds and found that all habitats examined allowed for successful larval development, survival and growth in the surrounding terrestrial environment are key to maintain and establish viable populations in these systems. Because many studies now indicate that anurans can use created and impacted aquatic habitats successfully [5, 9, 13, 14, 20, 35, 46], understanding the role that land use and land change have on terrestrial life stages is a critical next step to determine the full potential these habitats offer to native amphibians.

## Acknowledgments

We thank A. Gordon, T. Hoskins, K. Inoue, and S. Rumschlag for assistance in the field and comments on earlier drafts of this manuscripts; R. Kolb at the Miami University Ecology Research Center with enclosure construction; B. Mette for nutrient analyses; and all the landowners who allowed us access to their property and use of their ponds throughout the summer.

## Author Contributions

**Conceptualization:** Melissa B. Youngquist, Michelle D. Boone.

**Data curation:** Melissa B. Youngquist.

**Formal analysis:** Melissa B. Youngquist.

**Funding acquisition:** Melissa B. Youngquist, Michelle D. Boone.

**Investigation:** Melissa B. Youngquist.

**Methodology:** Melissa B. Youngquist, Michelle D. Boone.

**Project administration:** Melissa B. Youngquist.

**Resources:** Melissa B. Youngquist, Michelle D. Boone.

**Software:** Melissa B. Youngquist.

**Supervision:** Melissa B. Youngquist.

**Validation:** Melissa B. Youngquist.

**Visualization:** Melissa B. Youngquist.

**Writing – original draft:** Melissa B. Youngquist.

**Writing – review & editing:** Melissa B. Youngquist, Michelle D. Boone.

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
