## [Decision Letter · Decision Letter 0]

14 Apr 2021

PONE-D-21-04128

Independent of surrounding terrestrial habitat, human-made ponds create valuable habitat for pond-breeding anurans in agricultural landscapes

PLOS ONE

Dear Dr. Boone,

Thank you for submitting your manuscript to PLOS ONE. After careful consideration, we feel that it has merit but does not fully meet PLOS ONE’s publication criteria as it currently stands. Therefore, we invite you to submit a revised version of the manuscript that addresses the points raised during the review process.

We look forward to receiving your revised manuscript.

Kind regards,

Ji-Zhong Wan

Academic Editor

PLOS ONE

Additional Editor Comments:

Please revise the manuscript according to the reviewers' comments fully.

Journal Requirements:

3. In your Methods section, please include a comment about the state of the animals following this research. Were they euthanized or housed for use in further research? If any animals were sacrificed by the authors, please include the method of euthanasia and describe any efforts that were undertaken to reduce animal suffering.

Reviewers' comments:

Reviewer's Responses to Questions

**Comments to the Author**

1. Is the manuscript technically sound, and do the data support the conclusions?

Reviewer #1: Yes

Reviewer #2: Yes

2. Has the statistical analysis been performed appropriately and rigorously? 

Reviewer #1: Yes

Reviewer #2: Yes

3. Have the authors made all data underlying the findings in their manuscript fully available?

Reviewer #1: No

Reviewer #2: Yes

4. Is the manuscript presented in an intelligible fashion and written in standard English?

Reviewer #1: Yes

Reviewer #2: Yes

5. Review Comments to the Author

Reviewer #1: The manuscript by Youngquist and Boone is a nice work showing that human-made ponds that exhibit great variation in phytoplankton biomass and that have a vegetation buffer are valuable habitats for pond-breeding amphibians, regardless of their terrestrial surroundings. Unfortunately, the small number of ponds analyzed restrict the conclusions to ponds subject to the specific conditions as those analyzed (with a surrounding buffer area and high environmental variability). These limitations, however, are all well addressed in the discussion. I just feel that some sentences in the conclusions should be a bit more linked to these limitations (see specific comments). It is hard to say from these results that human-made ponds in agricultural fields are as good as those in any other terrestrial matrix. But we can say that other factors might be more important (but we do not know why exactly they are varying here). That being said, this is still valuable work, done with great care. The authors seemed to have handled well all the unexpected difficulties inherent to any experimental approach, especially regarding field experiments. The whole paper is generally well written. The introduction builds logically to the hypotheses tested, and the discussion, as already said, acknowledges the limitations of the experimental design and draws conclusions based on the results. The methods and results are generally good but could use a bit more details on some aspects of the methodology and the variability in environmental conditions found between ponds. I believe the figures can also be improved. Furthermore, I believe all of these issues are addressable. See specific comments below:

Introduction

Lines 90 - 91 – What is the difference expected between the two species?

Methods

Line 115 – Is the city water treated with chlorine or any chemical compounds that could be harmful in any way or change the behavior of the tadpoles?

Lines 116 – 117 - Tadpoles were maintained in mesocosms for how much time until they were transferred to the in situ enclosures? I know the reader could do the math, but it's easier if you explicitly give the information here.

Line 129 - Is the presence of fish common among these constructed ponds? Are they introduced by landowners? Or do they colonize ponds naturally after some time?

Lines 134 – 135 - Do you have any information on what kind of fertilizers and pesticides are used in those different crops? More importantly, do you know when they are usually applied? Some crops heavily rely on insecticides while others rely more on herbicides and they may have quite different consequences to amphibians as shown in:

Relyea RA. The lethal impact of Roundup on aquatic and terrestrial amphibians. Ecological Applications. 2005;15: 1118–1124. doi:10.1890/04-1291

Relyea RA, Schoeppner NM, Hoverman JT. Pesticides and amphibians: The importance of community context. Ecological Applications. 2005;15: 1125–1134. doi:10.1890/04-0559

Lines 146 – 147. Do you have pictures of each of those ponds and the enclosures to add as supplements? I often like to see pictures to have a better idea of the study system.

Lines 152 – 155. Did these new tadpoles also enter the experiment in stage 25? From what habitat types were those enclosures? Were they all aggregated in a single habitat? Single pond? Were they evenly distributed across ponds and habitats? This is important.

Lines 156 – 158. Did you test the water for pesticides?

Lines 181 – 182. “we calculated percent survival and average mass, time to metamorphosis, and Gosner developmental stage”. You already said that at the beginning of the paragraph.

Lines 182 – 183. “Percent survival and mass for both species were transformed to meet the assumption of normality”. What kind of transformation?

Lines 189 – 191. Any guesses on the reason for these high levels of nutrients?

Lines 196 – 198 - I am worried that you might be losing some relevant information by averaging all your values over time. Did you look at the variances over time for each habitat type as well? Maybe some habitat types exhibit greater variation in water quality variables over time, and that may be detrimental to tadpoles. Agricultural fields usually receive pesticide and fertilizer pulses in specific moments of their crop cycles. This may cause water parameter values to change drastically and then return to regular levels. If that happened, it could not show up as clearly in the mean as it would in the variance.

Lines 202 – 204. If you tested these variables separately, you do not have to exclude any of them because of multicollinearity. Unless you want to avoid making too many statistical tests (reducing your overall statistical power).

Lines 215 – 216. Again, it would be nice to see pictures as supplements.

Lines 224 – 225 “Enclosures were checked weekly until no more animals were encountered. 225 (19 April 2014)”. I did not understand this. Why were they not encountered? Did they die? Even the ones that “survived” overwinter? Did they escape? Or do you mean the enclosures from the first experiment?

Lines 229 – 230. What statistical distribution (or family) did you use in your generalized linear models?

Lines 230 – 231. I believe it would best to use the terrestrial enclosure as a random effect in a mixed model approach (maybe even the aquatic one too but could make the model too complex for the sample size and it may not converge). You can do that in a generalized linear mixed model. Packages lme4 and glmmTMB do that in R.

Results

Line 238 – “(F < 2.21, P > 0.147)”. Aren’t those exact F and P values? If so (even if they were rounded to fewer decimal digits) you can use an = sign instead of < or >.

Lines 243 – 246. Even though it was not "truly" significant, I think you can also highlight this result in table 2 (Phytoplankton and bullfrog variables). Maybe in italic to differentiate from the ones that were significant even after correction.

Lines 271 – 273. It is odd that pond affected mass at metamorphosis, initial mass (after metamorphosis) affected overwinter survival, but pond did not affect overwinter survival. Do you know why? Poor statistical power perhaps?

Lines 274. “final mass (F1,46 = 12.82, P = 0.0008), and change in mass (F1,46 = 4.20, P = 0.0462)”. Why not show this in figure 4 as well?

Figures. For all figures. The font size of the axis labels is too small, maybe in a formatted paper, they will be difficult to read.

Figure 1. The way results are being shown here is odd. I understand the authors wanted to show ponds individually, but, it REALLY looks like there are three treatments (white, gray, and black) that are very different from each other. But this treatment does not exist. It is nearly impossible to look at the figure and not to think about that. On the other hand, it is not clear to me what would be the best solution for this. You could mix the colors a bit more, so black, gray, and white symbols do not stay together at the top or bottom of the graph. You could try to use different symbols for each pond (diamonds, stars, crosses, etc). Another option would be to apply different shades of gray that correspond to some important water variable, like Chlorophyll-a.

Discussion

Lines - 345 – 347 – “The lack of an overall pond effect, despite differences in size at metamorphosis between ponds, was surprising. However, there was overlap in the distribution of initial metamorph sizes among ponds.”. Indeed! I think this deserves more elaboration.

Lines – 362. “Within the Midwest”. What Midwest? I guess it is the Midwest of the USA, but it may not be clear to all readers.

Lines 363 - 366. Even though it has been discussed above and the information is present below, I think it should also be acknowledged here that this conclusion may be conditioned to some conditions, such as the presence of a vegetation buffer and great variability in resource availability caused by other factors. I believe you could rewrite the paragraph to strengthen the link between the conclusion and their restrictions.

Reviewer #2: The authors present a field study assessing the influence of terrestrial land-use on larval survival of two amphibian species. Albeit there are some potential sample size issues and variability within land-use type that may hinder their ability to draw strong conclusions, the authors acknowledged these issues and made stronger inferences, where possible. The authors conclude that the best predictor of larval survival was phytoplankton abundance, regardless of surrounding habitat type. This paper is well-written and will be a nice contribution to the field of amphibian conservation.

General comments:

The present study was designed, in part, to understand whether aquatic environments in human dominated landscapes might be more beneficial to a species that is thriving (the bullfrog) compared to a species with a shrinking range (cricket frog). However, this question is left unanswered. As the data collected here did not answer that question, it would be of interest for the authors to speculate in the Discussion what the answer might be or the next steps to uncover it.

In the Discussion, there are several statements that indicate the authors were surprised by the outcome of this study (Lines 297 363), yet data collected by them previously indicated cricket frog presence was not uncommon in agricultural ponds (from personal observations), are able to disperse across cropland (citation 63), and the presence of cricket frogs are not predicted by the surrounding landscape (citation 46). A deeper delve into what the differences between those studies and conclusions are vs. those that have found differences (citations 43, 44) is warranted.

The authors suggest that these created ponds can be “valuable” larval habitat for amphibians. Is there any evidence these ponds are serving as population sinks?

Given the finding that phytoplankton is the strongest predictor of tadpole success, a more thorough discussion of what controls phytoplankton abundance would be useful.

Specific comments:

Line 28: lowercase cricket frog. Check throughout.

Lines 127-128 indicate some ponds were used for irrigation. Were there any drawdowns or fluctuations in water levels?

Lines 136: How far were the grassland ponds from agricultural plots? How large are typical tracts in the CRP? Is it likely that there are still agricultural plots draining into these ponds indicating the potential for contaminant exposure regardless of immediate adjacent habitat?

Indicate the number of cricket frog tadpoles that survived but did not metamorphose.

Lines 182-183. Part of the usefulness of generalized linear mixed models is the use of link functions. This negates the need to transform data by using the appropriate distribution, recognizing the non-Gaussian nature of some datasets. For count data such as percent survival, the log-link function should be used.

Line 227: remove comma between overwinter survival and terrestrial growth. Replace with an “and”.

Line 228: Move “Terrestrial enclosures were the experimental unit” to the end of the following sentence describing the mixed model.

Line 238: Replace “was” with “were”

Lines 297-305. This paragraph feels like a lot of hedging and leaves the reader wondering what the value of this study is. Are there major flaws in the design of this study (inadequate sample size, incorrect labeling of habitat type)? Rework this paragraph to have less “howevers”. Combine this paragraph with the next paragraph so the discussion of issues surrounding the defined habitat types in this study are included in one paragraph that ends with the statements regarding what can be concluded with this study- that “created ponds in different land use contexts can provide suitable larval habitat”.

Lines 320: lowercase bullfrog. Check throughout.

Line 323: insert “that” between studies and have.

Line 348: insert “than” after less.

Line 350: What habitats enable tadpoles to reach large sizes at metamorphosis?

Figures and Tables

Despite the caption indicating otherwise, it is very hard not to try to see patterns among shapes in the figures. Try switching so that the shading is representative of habitat type and choose more than three shapes to represent individual ponds so that there are not the same three shapes between any two habitat types. This way the habitat type will still be represented by shading across figures and readers will not attempt to identify meaningless patterns.

Include description regarding shading and shapes in captions for Figures 1-3 so that the description is the same and readers don’t have to look back to understand the significance.

Table 1 also includes the results from the GLMMs, correct?

6. PLOS authors have the option to publish the peer review history of their article (what does this mean?). If published, this will include your full peer review and any attached files.

Reviewer #1: No

Reviewer #2: No

---

## [Author Response · Author response to Decision Letter 0]

14 May 2021

Please see attached file "Response To Reviewers," which details how we addressed each recommendation.

---

## [Decision Letter · Decision Letter 1]

18 Jun 2021

PONE-D-21-04128R1

Larval development and survival of pond-breeding anurans in an agricultural landscape impacted more by phytoplankton than surrounding habitat

PLOS ONE

Dear Dr. Boone,

Thank you for submitting your manuscript to PLOS ONE. After careful consideration, we feel that it has merit but does not fully meet PLOS ONE’s publication criteria as it currently stands. Therefore, we invite you to submit a revised version of the manuscript that addresses the points raised during the review process.

We look forward to receiving your revised manuscript.

Kind regards,

Ji-Zhong Wan

Academic Editor

PLOS ONE

Journal Requirements:

Additional Editor Comments:

Minor corrections should be finished.

Reviewers' comments:

Reviewer's Responses to Questions

**Comments to the Author**

1. If the authors have adequately addressed your comments raised in a previous round of review and you feel that this manuscript is now acceptable for publication, you may indicate that here to bypass the “Comments to the Author” section, enter your conflict of interest statement in the “Confidential to Editor” section, and submit your "Accept" recommendation.

Reviewer #1: All comments have been addressed

Reviewer #2: All comments have been addressed

2. Is the manuscript technically sound, and do the data support the conclusions?

Reviewer #1: Yes

Reviewer #2: Yes

3. Has the statistical analysis been performed appropriately and rigorously? 

Reviewer #1: Yes

Reviewer #2: Yes

4. Have the authors made all data underlying the findings in their manuscript fully available?

Reviewer #1: Yes

Reviewer #2: Yes

5. Is the manuscript presented in an intelligible fashion and written in standard English?

Reviewer #1: Yes

Reviewer #2: Yes

6. Review Comments to the Author

Reviewer #1: The authors have made substantial improvements to the manuscript clarifying several aspects of the methodology. As I have already said, the whole paper is very well written, the introduction builds logically to the hypotheses, and the discussion addresses the limitations of the experimental design in a way that does not weaken the conclusions that they were able to draw from the results.

I only have a couple of minor comments. See below (lines refer to the “clean” version of the text):

Line 202 – When you say, “generalized linear mixed models”: Just for clarity, since you could not fit the model with a binomial distribution and transformed the data to meet assumptions of normality, you could either say that used "linear mixed models" or "generalized mixed models with gaussian distribution".

Line 215 – Replace "We used pond was a random effect" with "We used pond as a random effect".”

Lines 248 – 251: When you examine the effects of habitat type and pond origin on terrestrial growth and used the enclosure as the experimental unit, how exactly did get the values of your response variable? Did you average the final mass and change in mass within each enclosure?

Lines 252 – 23: When you examine the effects of initial mass on growth and survival, did you change the experimental unit from individuals to enclosures? Or did you keep individuals as the experimental unit as in the previous version? Please clarify that.

Reviewer #2: The authors thoughtfully addressed the previous concerns with this manuscript and the discussion tells a much more cohesive story. Well done! Below are a few minor grammatical edits with one additional point to address.

Comments:

Nearly half the animals didn’t metamorphose in one of the forested ponds, any indication what was different with that pond? The response of survival to metamorphosis- did this exclude the number of tadpoles collected that hadn’t metamorphosed or are they counted as “dead”?

Lines 208-211: Swap the order of results presented so it can read “one or two orders of magnitude, respectively”

Line 215: “was” should be “as”

Line 323: remove comma after “use”

Line 352: edit to “resource”

7. PLOS authors have the option to publish the peer review history of their article (what does this mean?). If published, this will include your full peer review and any attached files.

Reviewer #1: No

Reviewer #2: No

---

## [Editor Report · Decision Letter 2]

9 Jul 2021

Larval development and survival of pond-breeding anurans in an agricultural landscape impacted more by phytoplankton than surrounding habitat

PONE-D-21-04128R2

Dear Dr. Boone,

We’re pleased to inform you that your manuscript has been judged scientifically suitable for publication and will be formally accepted for publication once it meets all outstanding technical requirements.

Kind regards,

Ji-Zhong Wan

Academic Editor

PLOS ONE
---

## [Editor Report · Acceptance letter]

16 Jul 2021

PONE-D-21-04128R2 

Larval development and survival of pond-breeding anurans in an agricultural landscape impacted more by phytoplankton than surrounding habitat 

Dear Dr. Boone:

I'm pleased to inform you that your manuscript has been deemed suitable for publication in PLOS ONE. Congratulations! Your manuscript is now with our production department. 

Kind regards, 

on behalf of

Dr. Ji-Zhong Wan 

Academic Editor

PLOS ONE